# Non-stochastic Bandits With Evolving Observations

**Yogev Bar-On**                                                        BARONYOGEV@GMAIL.COM
*Tel Aviv University, Israel*

**Yishay Mansour**                                                 MANSOUR.YISHAY@GMAIL.COM
*Tel Aviv University and Google Research, Israel*

**Editors:** Gautam Kamath and Po-Ling Loh

## Abstract

We introduce a novel online learning framework that unifies and generalizes pre-established models, such as delayed and corrupted feedback, to encompass adversarial environments where action feedback evolves over time. In this setting, the observed loss is arbitrary and may not correlate with the true loss incurred, with each round updating previous observations adversarially. We propose regret minimization algorithms for both the full-information and bandit settings, with regret bounds quantified by the average feedback accuracy relative to the true loss. Our algorithms match the known regret bounds across many special cases, while also introducing previously unknown bounds.

**Keywords:** online learning, multi-armed bandits, delayed feedback

## 1. Introduction

In many sequential decision problems, the outcomes of actions are not immediately observed, and rather could only be estimated. Those estimations may be changing constantly (usually becoming more accurate) until certainty about the outcome is achieved.

This situation is common in financial settings, such as options trading, where the underlying asset price at the time of exercising can only be estimated at the time of the trade. Another more modern example is trading on blockchain systems (see, e.g., Bar-On and Mansour, 2023), where new blocks are created immediately but may be deleted with a low probability after a certain time (due to "forking", as in Neudecker and Hartenstein, 2019).

Another application is online advertising, where the value of an ad may evolve over time. For example, a user's click indicates some positive value. If the user completes an online form at a later time, the value increases. The value can then increase again if the user completes a purchase, or decrease if the user leaves without further action.

While working under these conditions, the theory of online learning with delayed feedback (Cesa-Bianchi et al., 2016; Thune et al., 2019) is useful for making informed decisions. However, there is still a big gap between theory and practice - primarily since the estimations of action values are not considered before the true value becomes known.

**Evolving feedback**    To bridge this gap, we propose a new framework for online decision-making in which the feedback on actions taken by the agent evolves and changes over time. Notably, the feedback can change retroactively, overriding previous observations. This applies to and generalizes various established feedback mechanisms. Those include delayed feedback, where all the information about the loss (the action's result) is revealed at a future time; composite feedback, where

the loss is revealed monotonically over time; and corrupted feedback, where the true loss is never revealed.

We investigate online learning environments with $K$ actions over $T$ rounds. An oblivious adversary chooses ahead of time not only the true loss $\ell_t \in [0,1]^K$ for round $t$, but also the *feedback loss* $\ell_\tau^{(t)} \in [0,1]^K$, which represents the new observation regarding the actions taken previously at any step $\tau \leq t$, overriding any previous observations.

As usual, at each round $t$ the agent chooses an action $a_t \in [K]$. The observations do not affect the suffered loss, and our objective is still to minimize the expected regret $R(T)$ in comparison to the best true loss in hindsight:

$$R(T) \triangleq \max_{a \in [K]} \mathbb{E} \left[ \sum_{t=1}^{T} \ell_{t,a_t} - \ell_{t,a} \right].$$

In our model, the observed losses may or may not correlate with the true loss. In the latter case, the agent gains no information and thus no strategy can guarantee a low regret. Fortunately, in real-life situations we expect the observations to be good estimates of the true loss and be more accurate as time progresses. Hence our algorithms' regret bound is smaller the more accurate the feedback is, and depends on the accuracy term

$$\Lambda = \sum_{t=1}^{T} \sum_{\tau=1}^{t} \min \left\{ 1, \left\| \ell_\tau - \ell_\tau^{(t)} \right\|_2 \right\}.$$

## 1.1. Contributions and outline

**Full-information setting**  We start with presenting an Exponential Weights (Cesa-Bianchi et al., 1997; Cesa-Bianchi and Lugosi, 2006) variant for the full-information setting in Section 2, where all the feedback generated by the adversary is revealed to the agent, and show a regret bound better than $\widetilde{O}\left(\sqrt{T + \Lambda}\right)$, where $\widetilde{O}$ hides logarithmic terms.

**Bandit setting**  We then present a Follow-The-Regularized-Leader (FTRL) (Abernethy et al., 2008; Orabona, 2019) variant for the bandit setting in Section 3, where only the feedback generated for actions taken by the agent is revealed, and show a regret bound of $\widetilde{O}\left(\sqrt{KT + \Lambda}\right)$. Our novel analysis creatively adapts methods from the delayed setting, where we quantify the information revealed at each step using the feedback accuracy, instead of a binary revealed/not revealed.

We can consider the delayed setting as a special case, where if the delay of round $\tau$ is $d_\tau$, then $\ell_\tau^{(t)} = \ell_\tau$ for any $t \geq \tau + d_\tau$, and $\ell_\tau^{(t)} = 0$ otherwise. Hence, $\Lambda = \sum_{t=1}^{T} d_t$ captures the total delay. Thus, our regret bounds are optimal (up to logarithmic terms) when used in the delayed setting (Cesa-Bianchi et al., 2016).

**Applications**  We show more special cases in Section 4. A key benefit to our framework is that it naturally supports any combination of those applications, for example, delayed feedback that is sometimes corrupted.

- In the optimistic delayed feedback environment (Flaspohler et al., 2021; Hsieh et al., 2022), where hints on delayed feedback are available to the agent, we match the existing regret bound in the full-information setting and show the *first regret bound* in the bandit setting.

- In the corrupted feedback environment (Resler and Mansour, 2019; Hajiesmaili et al., 2020), where the true losses are never revealed, we get a standard $\widetilde{O}\left(\sqrt{KT} + \mathcal{C}\right)$ bound, where $\mathcal{C}$ is the corruption budget.

- In the composite delayed feedback environment (Cesa-Bianchi et al., 2018; Wang et al., 2021), where the feedback is spread over $d$ partial consecutive observations, we apply our framework to get the optimal $\widetilde{O}\left(\sqrt{(K + d)T}\right)$ regret bound.

For clarity, we defer detailed proofs to the appendix.

## 1.2. Additional related works

Online learning under adversarial delayed feedback has been studied extensively both under the full-information (Weinberger and Ordentlich, 2002; Joulani et al., 2013, 2016) and bandit (Bistritz et al., 2019; Zimmert and Seldin, 2020; Ito et al., 2020; Gyorgy and Joulani, 2021; Jin et al., 2022; Van Der Hoeven and Cesa-Bianchi, 2022; Li and Guo, 2023) settings. Our analysis in this work is inspired by a recent work (van der Hoeven et al., 2023) that unifies the analysis of delayed feedback under many regimes such as linear bandits and Markov decision processes.

Many works study stochastic delayed environments as well (Agarwal and Duchi, 2011; Vernade et al., 2020; Pike-Burke et al., 2018; Gael et al., 2020; Lancewicki et al., 2021; Tang et al., 2024), and there are optimal algorithms for both cases simultaneously ("best of both worlds") (Masoudian et al., 2022, 2023).

Also generalized by our work is a corrupted adversarial feedback environment, previously studied for the stochastic case as well (Lykouris et al., 2018; Amir et al., 2020; Ito, 2021; He et al., 2022).

## 2. Evolving Exponential Weights

We start by presenting a simple regret minimization algorithm for the full information setting, where after step $t$ the agent observes all the feedback losses $\ell_\tau^{(t)}$ for all $\tau \leq t$. Our proposed algorithm is a modified version of Exponential Weights (Cesa-Bianchi et al., 1997; Cesa-Bianchi and Lugosi, 2006), summarized in Algorithm 1. The idea is for the agent to continuously update their beliefs on the true loss, even retroactively, as more feedback is presented.

Specifically, at each step $t$, the agent maintains a probability distribution $p$ over the set of possible actions as a function of an estimated total loss $L \in \mathbb{R}_+^K$:

$$p_i(L) = \frac{e^{-\eta L_i}}{\sum_{j \in [K]} e^{-\eta L_j}}. \tag{1}$$

where $\eta$ is some learning rate chosen by the agent. In our case, the agent's estimation of the total loss is based on the most recently observed feedback losses:

$$L_t^{\mathrm{e}} = \sum_{\tau=1}^{t-1} \ell_\tau^{(t-1)}.$$

---

**Algorithm 1** Evolving Exponential Weights

---

**Input:** $K, T \in \mathbb{N}; \eta > 0$

$L_{1,i}^{\mathrm{e}} \leftarrow 0$ for all $i \in [K]$;

**for** $t \leftarrow 1$ **to** $T$ **do**

    Set $p_i\left(L_t^{\mathrm{e}}\right) \leftarrow \dfrac{e^{-\eta L_{t,i}^{\mathrm{e}}}}{\sum_{j \in [K]} e^{-\eta L_{t,j}^{\mathrm{e}}}}$ for all $i \in [K]$;

    Play a random action $a_t \sim p\left(L_t^{\mathrm{e}}\right)$ and observe $\ell_\tau^{(t)}$ for all $\tau \leq t$;

    Set $L_{t+1}^{\mathrm{e}} \leftarrow \sum_{\tau=1}^{t} \ell_\tau^{(t)}$;

**end**

---

To quantify the regret, we will use the *total feedback inaccuracy* $D$ of the adversary:

$$D \triangleq \sum_{t=1}^{T} \|L_t^{\mathrm{e}} - L_t\|_\infty,$$

denoting the true total loss up to step $t$ by $L_t = \sum_{\tau=1}^{t-1} \ell_\tau$. Note this term does not depend on the agent's actions, but only on the losses generated by the adversary. It captures the magnitude of the difference between the observed and true losses.

In the case of a delayed setting with delay $d_t$ we have that $\|L_t^{\mathrm{e}} - L_t\|_\infty \leq d_t$, and thus $D \leq \sum_{t=1}^{T} d_t$, generalizing the total delay term.

## 2.1. Analysis

To analyze the regret of Algorithm 1, we will start by separating the regret into an *observation drift term* and an *auxiliary regret* term, as usually done when analyzing delayed settings. We can present the expected regret, assuming $a^*$ is the optimal action, as:

$$
\begin{aligned}
R(T) &= \mathbb{E}\left[\sum_{t=1}^{T} (\ell_{t,a_t} - \ell_{t,a^*})\right] \\
&= \sum_{t=1}^{T} \left(p(L_t^{\mathrm{e}}) \cdot \ell_t - \ell_{t,a^*}\right) \\
&= \underbrace{\sum_{t=1}^{T} \left(p(L_t^{\mathrm{e}}) - p(L_t)\right) \cdot \ell_t}_{\text{observation drift}} + \underbrace{\sum_{t=1}^{T} \left(p(L_t) \cdot \ell_t - \ell_{t,a^*}\right)}_{\text{auxiliary regret}}.
\end{aligned}
\tag{2}
$$

A standard Exponential Weights analysis bounds the auxiliary regret:

**Lemma 1** *Computing $p$ as in Eq. (1), we have for any action $a \in [K]$:*

$$\sum_{t=1}^{T} (p(L_t) \cdot \ell_t - \ell_{t,a}) \leq \frac{\ln K}{\eta} + \frac{\eta}{2}T.$$

To bound the drift term, we will use the following lemma:

**Lemma 2** *Let $\ell \in [0,1]^K$ be some loss vector, and $L_1, L_2$ two different estimations of the total loss. If the probability $p$ is computed as described in Eq. (1), we get:*

$$(p(L_1) - p(L_2)) \cdot \ell \leq 2\eta \|L_1 - L_2\|_\infty.$$

By substituting the results of Lemmas 1 and 2 in Eq. (2), we can now derive the main result for this section:

**Theorem 3** *The expected regret of Algorithm 1 after $T$ rounds holds:*

$$R(T) \leq \frac{\ln K}{\eta} + \eta \sum_{t=1}^{T} \left( \frac{1}{2} + 2\|L_t^e - L_t\|_\infty \right) = \frac{\ln K}{\eta} + \eta \left( \frac{T}{2} + 2D \right).$$

Optimally, we would like to set $\eta = \sqrt{\frac{\ln K}{\frac{T}{2} + 2D}}$, but we do not necessarily know the value of $D$ ahead of time. We can either use a standard doubling trick (Bistritz et al., 2019; Lancewicki et al., 2022) to get a parameter-independent algorithm, or use a known upper bound:

**Corollary 4** *Let $\bar{D}$ be a known upper bound on the feedback inaccuracy, such that:*

$$\bar{D} \geq D = \sum_{t=1}^{T} \|L_t^e - L_t\|_\infty.$$

*Using $\eta = \sqrt{\frac{\ln K}{\frac{T}{2} + 2\bar{D}}}$, the expected regret of Algorithm 1 after $T$ rounds holds:*

$$R(T) \leq \sqrt{4\ln K \left( \frac{T}{2} + 2\bar{D} \right)}.$$

In the delayed setting where $D$ is the total delay, this bound is known to be optimal (Cesa-Bianchi et al., 2016). Note that in the worst case, $D = \Omega\left(T^2\right)$. This implies that if the adversary provides no feedback until time $T$ on any of the losses, the regret is potentially $\Theta(T)$, as should be expected in such a scenario.

## 3. Evolving FTRL

In the bandit setting, the agent only receives feedback losses for actions they played at the times they were played. So if at time $\tau$ that agent chose action $a_\tau$, it will now receive $\ell_{\tau,a_\tau}^{(t)}$ for all $t \geq \tau$, and will *not* receive $\ell_{\tau,a}^{(t)}$ for any other action $a \neq a_\tau$.

We will use a Follow-The-Regularized-Leader (FTRL) (Abernethy et al., 2008; Orabona, 2019) variant as a strategy, presented in Algorithm 2. Given an estimated total loss $L \in \mathbb{R}_+^K$, we compute the probabilities over the set of actions as follows:

$$p(L) = \underset{p \in \Delta_{K-1}}{\arg\min} \left( p \cdot L + \Phi(p) \right), \tag{3}$$

where $\Delta_{K-1}$ is the $K$-dimensional simplex and $\Phi$ is the *regularization function*. Specifically in our analysis, we will use a standard negative entropy with log barrier regularization:

$$\Phi_{\eta,\gamma}(p) \triangleq \sum_{i \in [K]} \left( \frac{p_i}{\eta} - \frac{1}{\gamma} \right) \ln p_i \tag{4}$$

for some parameters $\eta, \gamma > 0$.

**Loss estimates**  Similar to what we did in the full-information case, as an estimation for the total loss at step $t$ we will use the most recent update:

$$\widehat{L}_t^{\mathrm{e}} = \sum_{\tau=1}^{t-1} \hat{\ell}_\tau^{(t-1)},$$

where $\hat{\ell}_\tau^{(t-1)}$ is an unbiased loss estimate calculated as:

$$\hat{\ell}_{\tau,a}^{(t-1)} = \ell_{\tau,a}^{(t-1)} \frac{\mathbb{1}\left[a = a_\tau\right]}{p_{a_\tau}\left(\widehat{L}_\tau^{\mathrm{e}}\right)}$$

for all $\tau < t$ and $a \in [K]$. Although these definitions might appear circular at first, we highlight that to compute $\widehat{L}_t^{\mathrm{e}}$ there is a need only for prior values of $\widehat{L}_\tau^{\mathrm{e}}$ where $\tau < t$.

Useful for our analysis is also $\hat{\ell}_\tau$, the unbiased loss estimate when the feedback is accurate:

$$\hat{\ell}_{\tau,a} = \ell_{\tau,a} \frac{\mathbb{1}\left[a = a_\tau\right]}{p_{a_\tau}\left(\widehat{L}_\tau^{\mathrm{e}}\right)}.$$

**Feedback accuracy measure**  In contrast to the full-information setting, we will quantify the feedback accuracy differently, using what we call the *feedback inaccuracy coefficients*:

$$\lambda_\tau^{(t)} \triangleq \frac{\left\|\ell_\tau^{(t)} - \ell_\tau\right\|_2}{1 + \left\|\ell_\tau^{(t)} - \ell_\tau\right\|_2} = \Theta\left(\min\left\{1, \left\|\ell_\tau^{(t)} - \ell_\tau\right\|_2\right\}\right),$$

measuring the feedback inaccuracy at step $t$ about the losses of step $\tau$. We again emphasize that the value of $\lambda_\tau^{(t)}$ does not depend on the agent's actions.

We will also denote by $\lambda_t = \sum_{\tau=1}^{t-1} \lambda_\tau^{(t-1)}$ the total feedback inaccuracy measure at step $t$, and by $d_{\max} = \max\left\{d \mid \exists_t \left(\ell_{t-d}^{(t)} \neq \ell_{t-d}\right)\right\}$ the maximal amount of rounds that the feedback can evolve.

Again taking an example from the delayed setting, we can see that $\lambda_t \leq d_t$ and thus the accuracy measures generalize the delays.

---

**Algorithm 2** Evolving FTRL

---

**Input:** Function $\Phi$; $K, T \in \mathbb{N}$

$\widehat{L}_{1,i}^{\mathrm{e}} \leftarrow 0$ for all $i \in [K]$;

**for** $t \leftarrow 1$ **to** $T$ **do**

  Set $p\left(\widehat{L}_t^{\mathrm{e}}\right) \leftarrow \arg\min_{p \in \Delta_{K-1}} \left(p \cdot \widehat{L}_t^{\mathrm{e}} + \Phi\left(p\right)\right)$;

  Play a random action $a_t \sim p\left(\widehat{L}_t^{\mathrm{e}}\right)$ and observe $\ell_{\tau,a}^{(t)}$ for all $\tau \leq t$ such that $a = a_\tau$;

  Set $\hat{\ell}_{\tau,a}^{(t)} \leftarrow \ell_{\tau,a}^{(t)} \frac{\mathbb{1}[a=a_\tau]}{p_{a_\tau}(\widehat{L}_\tau^{\mathrm{e}})}$ for all $\tau \leq t$ and $a \in [K]$;

  Set $\widehat{L}_{t+1}^{\mathrm{e}} \leftarrow \sum_{\tau=1}^{t} \hat{\ell}_\tau^{(t)}$;

**end**

---

## 3.1. Analysis

For the analysis of Algorithm 2, we will follow a method similar to (van der Hoeven et al., 2023). We will separate the regret into different drift terms, bounding each independently. Our novelty comes from new intermediate loss estimates, parameterized by the feedback accuracy coefficients $\lambda_\tau^{(t)}$. Namely, we will use the following intermediate loss estimates:

$$\widetilde{L}_t^e = \sum_{\tau=1}^{t-1} \left( \left( 1 - \lambda_\tau^{(t-1)} \right) \hat{\ell}_\tau^{(t-1)} + \lambda_\tau^{(t-1)} \ell_\tau^{(t-1)} \right),$$

$$\widetilde{L}_t = \sum_{\tau=1}^{t-1} \left( \left( 1 - \lambda_\tau^{(t-1)} \right) \hat{\ell}_\tau + \lambda_\tau^{(t-1)} \ell_\tau \right),$$

$$\widehat{L}_t = \sum_{\tau=1}^{t-1} \hat{\ell}_\tau, \qquad \text{and} \qquad \widehat{L}_t^* = \widehat{L}_t + \hat{\ell}_t = \sum_{\tau=1}^{t} \hat{\ell}_\tau.$$

### 3.1.1. DRIFT TERMS

We can now represent the regret as (again denoting the optimal action by $a^*$):

$$\begin{aligned}
R(T) &= \mathbb{E} \left[ \sum_{t=1}^{T} (\ell_{t,a_t} - \ell_{t,a^*}) \right] \\
&= \sum_{t=1}^{T} \mathbb{E} \left[ p\left( \widehat{L}_t^e \right) \cdot \ell_t - \ell_{t,a^*} \right] \\
&= \sum_{t=1}^{T} \mathbb{E} \left[ \left( p\left( \widehat{L}_t^e \right) - p\left( \widehat{L}_t \right) \right) \cdot \ell_t \right] + \sum_{t=1}^{T} \mathbb{E} \left[ p\left( \widehat{L}_t \right) \cdot \ell_t - \ell_{t,a^*} \right].
\end{aligned} \qquad (5)$$

Note that

$$\mathbb{E}\left[ \mathbb{1}\left[ a = a_\tau \right] \mid a_0, \ldots, a_{\tau-1} \right] = \Pr\left[ a = a_\tau \mid a_0, \ldots, a_{\tau-1} \right] = p_{a_\tau}\left( \widehat{L}_\tau^e \right),$$

and thus our loss estimate is indeed unbiased relative to the feedback losses:

$$\mathbb{E}\left[ \hat{\ell}_\tau^{(t-1)} \right] = \mathbb{E}\left[ \ell_{\tau,a}^{(t-1)} \frac{\mathbb{1}\left[ a = a_\tau \right]}{p_{a_\tau}\left( \widehat{L}_\tau^e \right)} \right] = \mathbb{E}\left[ \ell_{\tau,a}^{(t-1)} \frac{p_{a_\tau}\left( \widehat{L}_\tau^e \right)}{p_{a_\tau}\left( \widehat{L}_\tau^e \right)} \right] = \ell_\tau^{(t-1)}.$$

In the same way $\mathbb{E}\left[\hat{\ell}_\tau\right] = \ell_\tau$, and hence, continuing Eq. (5):

$$
\begin{aligned}
R(T) &= \sum_{t=1}^{T} \mathbb{E}\left[\left(p\left(\widehat{L}_t^{\mathrm{e}}\right) - p\left(\widehat{L}_t\right)\right) \cdot \ell_t\right] + \sum_{t=1}^{T} \mathbb{E}\left[p\left(\widehat{L}_t\right) \cdot \hat{\ell}_t - \hat{\ell}_{t,a^*}\right] \\
&= \underbrace{\sum_{t=1}^{T} \mathbb{E}\left[\left(p\left(\widehat{L}_t^{\mathrm{e}}\right) - p\left(\widetilde{L}_t^{\mathrm{e}}\right)\right) \cdot \ell_t\right]}_{H_1} + \underbrace{\sum_{t=1}^{T} \mathbb{E}\left[\left(p\left(\widetilde{L}_t^{\mathrm{e}}\right) - p\left(\widetilde{L}_t\right)\right) \cdot \ell_t\right]}_{H_2} \\
&\quad + \underbrace{\sum_{t=1}^{T} \mathbb{E}\left[\left(p\left(\widetilde{L}_t\right) - p\left(\widehat{L}_t\right)\right) \cdot \ell_t\right]}_{H_3} + \underbrace{\sum_{t=1}^{T} \mathbb{E}\left[\left(p\left(\widehat{L}_t\right) - p\left(\widehat{L}_t^*\right)\right) \cdot \hat{\ell}_t\right]}_{H_4} \\
&\quad + \underbrace{\mathbb{E}\left[\sum_{t=1}^{T} p\left(\widehat{L}_t^*\right) \cdot \hat{\ell}_t - \hat{\ell}_{t,a^*}\right]}_{\text{cheating regret}}.
\end{aligned}
\tag{6}
$$

### 3.1.2. BOUNDS

As we can see from the above equation, we have a cheating regret and four drift terms $H_1, H_2, H_3, H_4$. We bound the cheating regret in the following lemma:

**Lemma 5** *Computing $p$ as in Eq. (3) and using regularization $\Phi_{\eta,\gamma}$ as in Eq. (4) for some $\eta, \gamma > 0$, we get:*

$$
\mathbb{E}\left[\sum_{t=1}^{T} p\left(\widehat{L}_t^*\right) \cdot \hat{\ell}_t - \hat{\ell}_{t,a^*}\right] \leq 1 + \frac{K \ln T}{\gamma} + \frac{\ln K}{\eta}.
$$

For the drift terms, we use the following:

**Lemma 6** *Computing $p$ as in Eq. (3) and using regularization $\Phi_{\eta,\gamma}$ as in Eq. (4) for some $\eta, \gamma > 0$ such that $\frac{1}{\sqrt{\gamma}} \geq 128\left(1 + d_{\max}\right)$, we have for the drift terms in Eq. (6):*

$$
H_1, H_3 \leq 8\eta\left(KT + \sum_{t=1}^{T} \lambda_t\right),
$$

$$
H_2 \leq 24\eta \sum_{t=1}^{T} \lambda_t,
$$

$$
H_4 \leq 8\eta KT.
$$

Substituting the results of Lemmas 5 and 6 in Eq. (6), we get our main result:

**Theorem 7** *By using regularization function $\Phi_{\eta,\gamma}$ as in Eq. (4) for some $\eta, \gamma > 0$ such that $\frac{1}{\sqrt{\gamma}} \geq 128\left(1 + d_{\max}\right)$, the expected regret of Algorithm 2 after $T$ rounds holds:*

$$
R(T) \leq 1 + \frac{K \ln T}{\gamma} + \frac{\ln K}{\eta} + \eta\left(24KT + 40 \sum_{t=1}^{T} \lambda_t\right).
$$

Same as in the full-information case, we can either use a doubling trick or use a known bound on the feedback accuracy as follows.

**Corollary 8** *Let* $\bar{\Lambda} \geq \sum_{t=1}^{T} \lambda_t$ *be a known upper bound on the total inaccuracy. Choosing* $\eta = \frac{1}{\sqrt{KT+\bar{\Lambda}}}$ *and* $\gamma = \eta K$, *the expected regret of Algorithm 2 using regularization* $\Phi_{\eta,\gamma}$ *as in Eq.* (4) *holds for any* $T \geq 256 K d_{\max}^4$:

$$R(T) = \widetilde{O}\left(\sqrt{KT + \bar{\Lambda}}\right).$$

Since in the delayed setting the total delay $\sum_{t=1}^{T} d_t \geq \sum_{t=1}^{T} \lambda_t$, we again capture the optimal asymptotic bound (up to logarithmic terms).

### 3.2. Skipping technique

Note that in cases where the maximal delay $d_{\max}$ is very large or unbounded, we cannot use Corollary 8 directly.

To accommodate this issue, we will use a skipping wrapper similar to the one used in delayed settings with unbounded delays (Thune et al., 2019; Zimmert and Seldin, 2020), presented in Algorithm 3. The idea is to wrap our regret minimization algorithm to receive new observations only up to a certain delay.

---
**Algorithm 3** Skipping wrapper
---
**Input:** Algorithm $\mathcal{A}$; $T \in \mathbb{N}$; $d_{\max} > 0$
**for** $t \leftarrow 1$ **to** $T$ **do**
    Receive an action distribution $p_t^{\mathcal{A}}$ from algorithm $\mathcal{A}$;
    Play a random action $a_t \sim p_t^{\mathcal{A}}$;
    Feed back to $\mathcal{A}$ all observed $\ell_{\tau,i}^{(t)}$ such that $\tau \geq t - d_{\max}$, and all observed $\ell_{\tau,i}^{(\tau+d_{\max})}$ such that $\tau < t - d_{\max}$;
**end**

---

**Lemma 9** *Denote the regret of some algorithm* $\mathcal{A}$ *compared to a loss sequence* $\ell_1, \ldots, \ell_T$ *with maximal delay* $d_{\max}$ *as* $R_{d_{\max}}^{\mathcal{A}}\left(\{\ell_t\}_{1 \leq t \leq T}\right)$. *When using Algorithm 3 with* $\mathcal{A}$, *the expected regret holds:*

$$R(T) \leq R_{d_{\max}}^{\mathcal{A}}\left(\left\{\ell_t^{(t+d_{\max})}\right\}_{1 \leq t \leq T}\right) + 2\sum_{t=1}^{T}\left\|\ell_t - \ell_t^{(t+d_{\max})}\right\|_{\infty}.$$

We thus obtain a small regret bound when the estimation accuracy improves with time and becomes very accurate for large delays.

**Corollary 10** *Denote* $d_{\max} = \lfloor \frac{1}{4}\left(\frac{T}{K}\right)^{\frac{1}{4}} \rfloor$. *Using Algorithm 2 wrapped in Algorithm 3 with* $\eta = \frac{1}{\sqrt{KT+\bar{\Lambda}}}$ *and* $\gamma = \eta K$, *let* $\bar{\Lambda} \geq \sum_{t=1}^{T} \lambda_t$ *be a known upper bound on the total inaccuracy of the feedback recieved from the wrapper. Assuming* $\left\|\ell_\tau - \ell_\tau^{(t)}\right\|_{\infty} \leq \varepsilon$ *for any* $t \geq \tau + d_{\max}$, *the expected regret holds for any* $T$:

$$R(T) = \widetilde{O}\left(\sqrt{KT + \bar{\Lambda}} + \varepsilon T\right).$$

## 4. Applications

Our framework generalizes many established online learning environments, some of which we present here.

### 4.1. Optimistic delayed feedback

Previously investigated in (Flaspohler et al., 2021; Hsieh et al., 2022) for the full-information setting is the optimistic delayed framework. In this model, the feedback is delayed by $d$ steps, but the agent has access to hints about it after choosing the action.

Specifically, at time $t$ the agent receives a hint $\tilde{\ell}_t \in [0, 1]^K$ that estimates $\ell_t$, before observing $\ell_t$ at time $t + d$. Applying it to our framework, we can define $\ell_\tau^{(t)} = \tilde{\ell}_\tau$ for any $\tau \leq t < \tau + d$, and $\ell_\tau^{(t)} = \ell_\tau$ otherwise.

Using Corollaries 4 and 8, we then obtain a full-information regret bound similar to (Flaspohler et al., 2021), and a *newly established* regret bound for the bandit setting.

**Corollary 11** *Using Algorithm 1 with optimistic delayed feedback in the full-information setting guarantees an expected regret of:*

$$\widetilde{O}\left(\sqrt{\sum_{t=1}^{T}\left\|\sum_{\tau=t-d+1}^{t}\left(\ell_\tau - \tilde{\ell}_\tau\right)\right\|_\infty}\right).$$

*In the bandit setting, using Algorithm 2 guarantees an expected regret of:*

$$\widetilde{O}\left(\sqrt{\sum_{t=1}^{T}\sum_{\tau=t-d+1}^{t}\min\left\{1, \left\|\ell_\tau - \tilde{\ell}_\tau\right\|_2\right\}}\right).$$

### 4.2. Corrupted feedback

In the corrupted feedback setting (Resler and Mansour, 2019; Hajiesmaili et al., 2020), true losses are never revealed, only some corrupted loss $\tilde{\ell}_t \in [0, 1]^K$ that is observed immediately. In terms of the evolving feedback framework, this is equivalent to having $\ell_\tau^{(t)} = \tilde{\ell}_\tau$ for any $\tau \leq t$.

To measure the amount of corruption, we denote the *corruption budget* by

$$\mathcal{C} \triangleq \sum_{t=1}^{T}\left\|\ell_t - \tilde{\ell}_t\right\|_\infty.$$

Since the true loss is never revealed and the maximal delay is infinite, we need only the result of the skipping technique (Lemma 9) to obtain a regret bound.

**Corollary 12** *Using Algorithm 3 with $d_{\max} = 0$ and any multi-armed bandit $\widetilde{O}\left(\sqrt{KT}\right)$ regret minimization algorithm, we obtain an*

$$\widetilde{O}\left(\sqrt{KT} + \mathcal{C}\right)$$

*expected regret in a corrupted environment.*

### 4.3. Composite delayed feedback

The composite delayed feedback setting is the case where each loss is spread into $d$ positive partial losses $\tilde{\ell}_t^{(1)}, \ldots, \tilde{\ell}_t^{(d)}$ that sum to $\ell_t$, observed consecutively by the agent. Applying it to our framework, we have that

$$\ell_\tau^{(t)} = \sum_{s=\tau+1}^{\min\{t+1, \tau+d\}} \tilde{\ell}_\tau^{(s-\tau)}.$$

Previous works (Cesa-Bianchi et al., 2018; Wang et al., 2021) discuss the case where the observations are *anonymous*. Namely, the agent observes only the sum of partial losses revealed in the current step. This fact does not generalize directly into our evolving feedback framework.

Hence, we will look at the non-anonymous scenario, where each observation can be attributed to a time and action. However, we can remove the limitation that the partial losses must be positive and can accommodate in our framework *negative* partial losses. The only restriction is that $\sum_{s=1}^{\bar{s}} \tilde{\ell}_t^{(s)} \in [0, 1]^K$ for any $1 \leq \bar{s} \leq d$.

We can thus obtain regret bounds using Corollaries 4 and 8.

**Corollary 13** *In a composite feedback environment, allowing negative partial losses, using Algorithm 1 in the full-information setting guarantees an expected regret of:*

$$\widetilde{O}\left(\sqrt{(1+d)T}\right).$$

*In the bandit setting, using Algorithm 2 guarantees an expected regret of:*

$$\widetilde{O}\left(\sqrt{(K+d)T}\right).$$

## 5. Discussion

This work introduces a framework for online learning under adversarial feedback that evolves over time. Our setting generalizes and unifies previously studied models like delayed, corrupted, and composite feedback.

We proposed regret minimization algorithms for both the full information (Algorithm 1) and the bandit (Algorithm 2) settings, achieving asymptotically optimal regret bounds (up to logarithmic terms) that depend on the average accuracy of the observed feedback compared to the true losses, using a novel analysis.

In addition to providing a unified model for many problems, our framework is beneficial for real-world scenarios such as finance and online advertising. By incorporating all available information on the value of actions, our approach achieves regret bounds that were previously not feasible.

Our work introduces a few follow-up research questions. Mainly, are our regret bounds optimal in terms of the instance-dependent average feedback accuracy? Currently, we can show optimality only in cases where the difference between the agent's estimations and the true loss is large, like in the delayed setting. It is an open question if our bounds could be improved for cases where the difference is small.

Another natural question to ask is whether we can expand our model to accommodate loss estimations for future rounds as well as past ones, and how the optimal regret bounds will behave in this scenario.

## Acknowledgments

This project has received funding from the European Research Council (ERC) under the European Union's Horizon 2020 research and innovation program (grant agreement No. 882396), by the Israel Science Foundation, the Yandex Initiative for Machine Learning at Tel Aviv University and a grant from the Tel Aviv University Center for AI and Data Science (TAD).

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

## Appendix A. Deferred proofs from Section 2

**Lemma 14** *Computing $p$ as in Eq. (1), we have for any action $a \in [K]$:*

$$\sum_{t=1}^{T} (p(L_t) \cdot \ell_t - \ell_{t,a}) \leq \frac{\ln K}{\eta} + \frac{\eta}{2}T.$$

**Proof** For any $t$, we have:

$$
\begin{aligned}
\frac{\sum_{j\in[K]} e^{-\eta L_{t+1,j}}}{\sum_{j\in[K]} e^{-\eta L_{t,j}}} &= \frac{\sum_{j\in[K]} e^{-\eta L_{t,j}} e^{-\eta \ell_{t,j}}}{\sum_{j\in[K]} e^{-\eta L_{t,j}}} \\
&= \sum_{j\in[K]} p(L_{t,j}) e^{-\eta \ell_{t,j}} \\
&\leq \sum_{j\in[K]} p(L_{t,j})(1 - \eta \ell_{t,j} + \frac{\eta^2}{2}\ell_{t,j}^2) \\
&\leq 1 - \eta\,(p(L_t) \cdot \ell_t) + \frac{\eta^2}{2},
\end{aligned}
$$

where we used the fact that $e^{-x} \leq 1 - x + \frac{x^2}{2}$ for $x \geq 0$.

For any $a \in [K]$, we thus have:

$$
\begin{aligned}
\frac{e^{-\eta L_{T,a}}}{K} &\leq \frac{\sum_{j\in[K]} e^{-\eta L_{T,j}}}{K} \\
&= \prod_{t=1}^{T} \frac{\sum_{j\in[K]} e^{-\eta L_{t+1,j}}}{\sum_{j\in[K]} e^{-\eta L_{t,j}}} \\
&\leq \prod_{t=1}^{T} \left(1 - \eta\,(p(L_t) \cdot \ell_t) + \frac{\eta^2}{2}\right).
\end{aligned}
$$

Taking logs of both sides and using the fact that $\ln(1 + x) \leq x$, we get the desired result. ∎

**Lemma 15** *Let $\ell \in [0,1]^K$ be some loss vector, and $L_1, L_2$ two different estimations of the total loss. If the probability $p$ is computed as described in Eq. (1), we get:*

$$(p(L_1) - p(L_2)) \cdot \ell \leq 2\eta \left\| L_1 - L_2 \right\|_\infty.$$

**Proof** We start by noting that:

$$(p(L_1) - p(L_2)) \cdot \ell = \sum_{i\in[K]} (p_i(L_1) - p_i(L_2))\,\ell_i = \sum_{i\in[K]} p_i(L_1) \left(1 - \frac{p_i(L_2)}{p_i(L_1)}\right) \ell_i. \qquad (7)$$

To bound this term, we will use the fact that:

$$\frac{p_i(L_2)}{p_i(L_1)} = \frac{e^{-\eta L_{2i}}}{e^{-\eta L_{1i}}} \frac{\sum_{j\in[K]} e^{-\eta L_{1j}}}{\sum_{j\in[K]} e^{-\eta L_{2j}}}$$

$$= e^{\eta(L_{1i}-L_{2i})} \frac{\sum_{j\in[K]} e^{-\eta L_{2j}-\eta(L_{1j}-L_{2j})}}{\sum_{j\in[K]} e^{-\eta L_{2j}}}$$

$$\geq e^{\eta(L_{1i}-L_{2i})-\eta \max_j (L_{1j}-L_{2j})}$$

$$\geq e^{\eta\left(\min_j (L_{1j}-L_{2j})-\max_j (L_{1j}-L_{2j})\right)}$$

$$\geq e^{-2\eta\|L_1-L_2\|_\infty}$$

$$\geq 1 - 2\eta \|L_1 - L_2\|_\infty .$$

Substituting in Eq. (7), we get:

$$(p(L_1) - p(L_2)) \cdot \ell$$
$$\leq \sum_{i\in[K]} 2p_i(L_1)\eta \|L_1 - L_2\|_\infty \ell_i$$
$$= 2\eta \|L_1 - L_2\|_\infty \sum_{i\in[K]} p_i(L_1)\ell_i$$
$$\leq 2\eta \|L_1 - L_2\|_\infty ,$$

as required. ∎

## Appendix B. Deferred proofs from Section 3

### B.1. Cheating regret bound

**Lemma 16** *Computing $p$ as in Eq. (3), for any fixed probability $q \in \Delta_{K-1}$ and regularization function $\Phi$:*

$$\sum_{t=1}^{T} p\left(\widehat{L}_t^*\right) \cdot \hat{\ell}_t + \Phi\left(p\left(\widehat{L}_1^*\right)\right) \leq \sum_{t=1}^{T} q \cdot \hat{\ell}_t + \Phi(q).$$

**Proof** We use a standard be-the-leader analysis and prove using induction on $T$. The base case $T = 1$ follows directly from Eq. (3), so we assume for any $q \in \Delta_{K-1}$:

$$\sum_{t=1}^{T-1} p\left(\widehat{L}_t^*\right) \cdot \hat{\ell}_t + \Phi\left(p\left(\widehat{L}_1^*\right)\right) \leq \sum_{t=1}^{T-1} q \cdot \hat{\ell}_t + \Phi(q).$$

Specifically, we can assign $q = p\left(\widehat{L}_T^*\right)$. Hence:

$$
\begin{aligned}
\sum_{t=1}^{T} p\left(\widehat{L}_t^*\right) \cdot \hat{\ell}_t + \Phi\left(p\left(\widehat{L}_1^*\right)\right) &= \sum_{t=1}^{T-1} p\left(\widehat{L}_t^*\right) \cdot \hat{\ell}_t + \Phi\left(p\left(\widehat{L}_1^*\right)\right) + p\left(\widehat{L}_T^*\right) \cdot \hat{\ell}_t \\
&\leq \sum_{t=1}^{T-1} p\left(\widehat{L}_T^*\right) \cdot \hat{\ell}_t + \Phi\left(p\left(\widehat{L}_T^*\right)\right) + p\left(\widehat{L}_T^*\right) \cdot \hat{\ell}_t \\
&= p\left(\widehat{L}_T^*\right) \cdot \widehat{L}_T^* + \Phi\left(p\left(\widehat{L}_T^*\right)\right) \\
&= \min_{q \in \Delta_{K-1}} \left(q \cdot \widehat{L}_T^* + \Phi(q)\right)
\end{aligned}
$$

as required, where the last equality is from Eq. (3). ∎

**Lemma 17** *Computing $p$ as in Eq. (3) and using regularization $\Phi_{\eta,\gamma}$ as in Eq. (4) for some $\eta, \gamma > 0$, we get:*

$$
\mathbb{E}\left[\sum_{t=1}^{T} p\left(\widehat{L}_t^*\right) \cdot \hat{\ell}_t - \hat{\ell}_{t,a^*}\right] \leq 1 + \frac{K \ln T}{\gamma} + \frac{\ln K}{\eta}.
$$

**Proof** Denote by $q^*$ the action probability that chooses $a^*$ with probability 1, and let

$$
q = \left(1 - \frac{1}{T}\right) q^* + \frac{1}{T} p\left(\widehat{L}_1^*\right).
$$

We get:

$$
\begin{aligned}
\sum_{t=1}^{T} \left(p\left(\widehat{L}_t^*\right) \cdot \hat{\ell}_t - \hat{\ell}_{t,a^*}\right) &= \sum_{t=1}^{T} \left(p\left(\widehat{L}_t^*\right) - q^*\right) \cdot \hat{\ell}_t \\
&= \sum_{t=1}^{T} \left(p\left(\widehat{L}_t^*\right) - q\right) \cdot \hat{\ell}_t + \sum_{t=1}^{T} (q - q^*) \cdot \hat{\ell}_t \\
&= \sum_{t=1}^{T} \left(p\left(\widehat{L}_t^*\right) - q\right) \cdot \hat{\ell}_t + \frac{1}{T} \sum_{t=1}^{T} \left(p\left(\widehat{L}_1^*\right) - q^*\right) \cdot \hat{\ell}_t \\
&\leq \sum_{t=1}^{T} \left(p\left(\widehat{L}_t^*\right) - q\right) \cdot \hat{\ell}_t + 1 \\
&\leq \Phi_{\eta,\gamma}(q) - \Phi_{\eta,\gamma}\left(p\left(\widehat{L}_1^*\right)\right) + 1,
\end{aligned}
$$

where we used Lemma 16 in the last step. Thus, using Eq. (4):

$$
\sum_{t=1}^{T} \left( p\left(\widehat{L}_t^*\right) \cdot \hat{\ell}_t - \hat{\ell}_{t,a^*} \right) \leq 1 + \sum_{i \in [K]} \left( \frac{q_i}{\eta} - \frac{1}{\gamma} \right) \ln q_i - \sum_{i \in [K]} \left( \frac{p_i\left(\widehat{L}_1^*\right)}{\eta} - \frac{1}{\gamma} \right) \ln p_i\left(\widehat{L}_1^*\right)
$$

$$
\leq 1 + \frac{1}{\gamma} \sum_{i \in [K]} \ln \frac{p_i\left(\widehat{L}_1^*\right)}{q_i} + \sum_{i \in [K]} \frac{p_i\left(\widehat{L}_1^*\right)}{\eta} \ln \frac{1}{p_i\left(\widehat{L}_1^*\right)}
$$

$$
= 1 + \frac{1}{\gamma} \sum_{i \in [K]} \ln \frac{p_i\left(\widehat{L}_1^*\right)}{\left(1 - \frac{1}{T}\right) q_i^* + \frac{1}{T} p_i\left(\widehat{L}_1^*\right)} + \sum_{i \in [K]} \frac{p_i\left(\widehat{L}_1^*\right)}{\eta} \ln \frac{1}{p_i\left(\widehat{L}_1^*\right)}
$$

$$
\leq 1 + \frac{K \ln T}{\gamma} + \frac{1}{\eta} \sum_{i \in [K]} p_i\left(\widehat{L}_1^*\right) \ln \frac{1}{p_i\left(\widehat{L}_1^*\right)}.
$$

Using Jensen's inequality, we can complete the proof:

$$
\sum_{t=1}^{T} \left( p\left(\widehat{L}_t^*\right) \cdot \hat{\ell}_t - \hat{\ell}_{t,a^*} \right) \leq 1 + \frac{K \ln T}{\gamma} + \frac{1}{\eta} \ln \sum_{i \in [K]} \frac{p_i\left(\widehat{L}_1^*\right)}{p_i\left(\widehat{L}_1^*\right)}
$$

$$
= 1 + \frac{K \ln T}{\gamma} + \frac{\ln K}{\eta}.
$$

∎

## B.2. Drift bounds preliminaries

We start with defining the dual norms on $x \in \mathbb{R}^K$ induced by a strictly-convex twice-differentiable regularization function $\Phi$ and a point $p \in \mathbb{R}^K$:

$$
\|x\|_{\Phi,p} \triangleq \sqrt{x^T \left(\nabla^2 \Phi\left(p\right)\right)^{-1} x} \qquad \text{and} \qquad \|x\|_{\Phi,p}^* \triangleq \sqrt{x^T \left(\nabla^2 \Phi\left(p\right)\right) x},
$$

where $\nabla^2 \Phi$ denotes the Hessian matrix of $\Phi$.

For $\Phi_{\eta,\gamma}$ as defined in Eq. (4) we get:

$$
\|x\|_{\Phi_{\eta,\gamma},p} = \sqrt{\sum_{i \in [K]} \frac{\eta \gamma p_i^2}{\eta + \gamma p_i} x_i^2} \qquad \text{and} \qquad \|x\|_{\Phi_{\eta,\gamma},p}^* = \sqrt{\sum_{i \in [K]} \frac{\eta + \gamma p_i}{\eta \gamma p_i^2} x_i^2}. \tag{8}
$$

For clarity, we will also denote the Dikin ellipsoid of radius $\frac{1}{2}$ as:

$$
\mathcal{D}_{\Phi}\left(p\right) \triangleq \left\{ x \in \mathbb{R}^K \mid \|x - p\|_{\Phi,p}^* \leq \frac{1}{2} \right\}.
$$

We will use the following facts (for proofs see Lemma 16, Lemma 1 and Lemma 9 in (van der Hoeven et al., 2023) respectively):

**Fact 1** *Let $x, p, q \in \mathbb{R}^K$. Using regularization $\Phi_{\eta,\gamma}$ as in Eq. (4) for some $\eta, \gamma > 0$, we get that if $q \in \mathcal{D}_{\Phi_{\eta,\gamma}}(p)$, then:*

$$\frac{1}{2}\|x\|_{\Phi_{\eta,\gamma},q} \leq \|x\|_{\Phi_{\eta,\gamma},p} \leq 2\|x\|_{\Phi_{\eta,\gamma},q}.$$

**Fact 2** *Let $L, L' \in \mathbb{R}_+^K$ and $q \in \mathbb{R}^K$. Computing $p$ as in Eq. (3) and using regularization $\Phi_{\eta,\gamma}$ as in Eq. (4) for some $\eta, \gamma > 0$, we get that if $p(L), p(L') \in \mathcal{D}_{\Phi_{\eta,\gamma}}(q)$, then:*

$$\left\|p(L') - p(L)\right\|_{\Phi_{\eta,\gamma},q}^* \leq 8\left\|L' - L\right\|_{\Phi_{\eta,\gamma},q}.$$

**Fact 3** *Let $L, L' \in \mathbb{R}_+^K$. Computing $p$ as in Eq. (3) and using regularization $\Phi_{\eta,\gamma}$ as in Eq. (4) for some $\eta, \gamma > 0$, we get that if $\|L' - L\|_{\Phi_{\eta,\gamma},p(L)} \leq \frac{1}{16}$, then:*

$$p(L') \in \mathcal{D}_{\Phi_{\eta,\gamma}}(p(L)).$$

### B.3. Drift bounds

**Lemma 18** *Computing $p$ as in Eq. (3) and using regularization $\Phi_{\eta,\gamma}$ as in Eq. (4) for some $\eta, \gamma > 0$ such that $\frac{1}{\sqrt{\gamma}} \geq 32d_{\max}$, we have for all $0 \leq d \leq d_{\max}$:*

$$p\left(\widehat{L}_{t+d}^{\mathrm{e}}\right) \in \mathcal{D}_{\Phi_{\eta,\gamma}}\left(p\left(\widehat{L}_t^{\mathrm{e}}\right)\right).$$

**Proof** We will prove by induction on $d$. The base case $d = 0$ is trivially true. We will thus assume the claim is true for any $d' < d$. Using Fact 3, we only need to show that:

$$\sum_{\tau=t}^{t+d-1} \left\|\hat{\ell}_\tau^{(t+d-1)} - \hat{\ell}_\tau^{(t-1)}\right\|_{\Phi_{\eta,\gamma},p\left(\widehat{L}_t^{\mathrm{e}}\right)} \leq \frac{1}{16}.$$

From our assumption, we can use Fact 1 and get:

$$\sum_{\tau=t}^{t+d-1} \left\|\hat{\ell}_\tau^{(t+d-1)} - \hat{\ell}_\tau^{(t-1)}\right\|_{\Phi_{\eta,\gamma},p\left(\widehat{L}_t^{\mathrm{e}}\right)} \leq 2\sum_{\tau=t}^{t+d-1} \left\|\hat{\ell}_\tau^{(t+d-1)} - \hat{\ell}_\tau^{(t-1)}\right\|_{\Phi_{\eta,\gamma},p\left(\widehat{L}_\tau^{\mathrm{e}}\right)}.$$

Hence, from Eq. (8):

$$\begin{aligned}
\left\|\hat{\ell}_\tau^{(t+d-1)} - \hat{\ell}_\tau^{(t-1)}\right\|_{\Phi_{\eta,\gamma},p\left(\widehat{L}_\tau^{\mathrm{e}}\right)}^2 &= \sum_{i\in[K]} \frac{\eta\gamma p_i^2\left(\widehat{L}_\tau^{\mathrm{e}}\right)}{\eta + \gamma p_i\left(\widehat{L}_\tau^{\mathrm{e}}\right)} \frac{\left(\ell_{\tau,i}^{(t+d-1)} - \ell_{\tau,i}^{(t-1)}\right)^2 \mathbb{1}\left[i = a_\tau\right]}{p_i^2\left(\widehat{L}_\tau^{\mathrm{e}}\right)} \\
&\leq \frac{\eta\gamma}{\eta + \gamma p_{a_\tau}\left(\widehat{L}_\tau^{\mathrm{e}}\right)} \\
&\leq \gamma,
\end{aligned}$$

and thus we get that

$$\sum_{\tau=t}^{t+d-1} \left\|\hat{\ell}_\tau^{(t+d-1)} - \hat{\ell}_\tau^{(t-1)}\right\|_{\Phi_{\eta,\gamma},p\left(\widehat{L}_t^{\mathrm{e}}\right)} \leq 2d\sqrt{\gamma} \leq 2d_{\max}\sqrt{\gamma} \leq \frac{1}{32}$$

as required. ∎

**Lemma 19** *Computing $p$ as in Eq. ([3](#)) and using regularization $\Phi_{\eta,\gamma}$ as in Eq. ([4](#)) for some $\eta, \gamma > 0$ such that $\frac{1}{\sqrt{\gamma}} \geq 128 \left(1 + d_{\max}\right)$, we have for the drift terms in Eq. ([6](#)):*

$$H_1, H_3 \leq 8\eta \left( KT + \sum_{t=1}^{T} \lambda_t \right),$$

$$H_2 \leq 24\eta \sum_{t=1}^{T} \lambda_t,$$

$$H_4 \leq 8\eta KT.$$

**Proof**

$H_1, H_3$.  We will prove the bound for $H_3$, and the proof for $H_1$ is identical. First, note that:

$$\widetilde{L}_t - \widehat{L}_t^{\mathrm{e}} = \sum_{\tau=1}^{t-1} \left( \left(1 - \lambda_\tau^{(t-1)}\right) \left(\hat{\ell}_\tau - \hat{\ell}_\tau^{(t-1)}\right) + \lambda_\tau^{(t-1)} \left(\ell_\tau - \hat{\ell}_\tau^{(t-1)}\right) \right).$$

Denote $t' = \max\{1, t - d_{\max}\}$. For any $\tau < t'$ we have that $\hat{\ell}_\tau = \hat{\ell}_\tau^{(t-1)}$ and $\lambda_\tau^{(t-1)} = 0$, and thus:

$$\left\| \widetilde{L}_t - \widehat{L}_t^{\mathrm{e}} \right\|_{\Phi_{\eta,\gamma},p\left(\widehat{L}_t^{\mathrm{e}}\right)} \leq \sum_{\tau=t'}^{t-1} \left\| \hat{\ell}_\tau - \hat{\ell}_\tau^{(t-1)} \right\|_{\Phi_{\eta,\gamma},p\left(\widehat{L}_t^{\mathrm{e}}\right)} + \sum_{\tau=t'}^{t-1} \left\| \ell_\tau - \hat{\ell}_\tau^{(t-1)} \right\|_{\Phi_{\eta,\gamma},p\left(\widehat{L}_t^{\mathrm{e}}\right)}$$

$$\leq \sum_{\tau=t'}^{t-1} \left\| \hat{\ell}_\tau \right\|_{\Phi_{\eta,\gamma},p\left(\widehat{L}_t^{\mathrm{e}}\right)} + \sum_{\tau=t'}^{t-1} \left\| \ell_\tau \right\|_{\Phi_{\eta,\gamma},p\left(\widehat{L}_t^{\mathrm{e}}\right)} + 2 \sum_{\tau=t'}^{t-1} \left\| \hat{\ell}_\tau^{(t-1)} \right\|_{\Phi_{\eta,\gamma},p\left(\widehat{L}_t^{\mathrm{e}}\right)}.$$

Using Fact [1](#) and Lemma [18](#) we can move to the norm induced by $p\left(\widehat{L}_\tau^{\mathrm{e}}\right)$:

$$\left\| \widetilde{L}_t - \widehat{L}_t^{\mathrm{e}} \right\|_{\Phi_{\eta,\gamma},p\left(\widehat{L}_t^{\mathrm{e}}\right)}$$

$$\leq 2 \sum_{\tau=t'}^{t-1} \left\| \hat{\ell}_\tau \right\|_{\Phi_{\eta,\gamma},p\left(\widehat{L}_\tau^{\mathrm{e}}\right)} + \sum_{\tau=t'}^{t-1} \left\| \ell_\tau \right\|_{\Phi_{\eta,\gamma},p\left(\widehat{L}_t^{\mathrm{e}}\right)} + 4 \sum_{\tau=t'}^{t-1} \left\| \hat{\ell}_\tau^{(t-1)} \right\|_{\Phi_{\eta,\gamma},p\left(\widehat{L}_\tau^{\mathrm{e}}\right)}.$$

We can now use Eq. ([8](#)) to see that

$$\left\| \hat{\ell}_\tau \right\|_{\Phi_{\eta,\gamma},p\left(\widehat{L}_\tau^{\mathrm{e}}\right)}^2 = \sum_{i \in [K]} \frac{\eta \gamma p_i^2 \left(\widehat{L}_\tau^{\mathrm{e}}\right)}{\eta + \gamma p_i \left(\widehat{L}_\tau^{\mathrm{e}}\right)} \frac{\ell_\tau^2 \mathbb{1}\left[i = a_\tau\right]}{p_i^2 \left(\widehat{L}_\tau^{\mathrm{e}}\right)}$$

$$\leq \frac{\eta \gamma}{\eta + \gamma p_{a_\tau} \left(\widehat{L}_\tau^{\mathrm{e}}\right)}$$

$$\leq \gamma,$$

and the same is true for $\left\| \hat{\ell}_\tau^{(t-1)} \right\|_{\Phi_{\eta,\gamma},p\left(\widehat{L}_\tau^{\mathrm{e}}\right)}^2$. Also:

$$\left\| \ell_\tau \right\|_{\Phi_{\eta,\gamma},p\left(\widehat{L}_t^{\mathrm{e}}\right)}^2 = \sum_{i \in [K]} \frac{\eta \gamma p_i^2 \left(\widehat{L}_t^{\mathrm{e}}\right)}{\eta + \gamma p_i \left(\widehat{L}_t^{\mathrm{e}}\right)} \ell_{\tau,i}^2 \leq \sum_{i \in [K]} \gamma p_i^2 \left(\widehat{L}_t^{\mathrm{e}}\right) \leq \gamma$$

as well. Hence:

$$\left\|\widetilde{L}_t - \widehat{L}_t^{\mathrm{e}}\right\|_{\Phi_{\eta,\gamma},p(\widehat{L}_t^{\mathrm{e}})} \leq 7d_{\max}\sqrt{\gamma} \leq \frac{1}{16}.$$

Similarly:

$$
\begin{aligned}
\left\|\widehat{L}_t - \widehat{L}_t^{\mathrm{e}}\right\|_{\Phi_{\eta,\gamma},p(\widehat{L}_t^{\mathrm{e}})} &\leq \sum_{\tau=1}^{t-1} \left\|\hat{\ell}_\tau - \hat{\ell}_\tau^{(t-1)}\right\|_{\Phi_{\eta,\gamma},p(\widehat{L}_t^{\mathrm{e}})} \\
&= \sum_{\tau=t'}^{t-1} \left\|\hat{\ell}_\tau - \hat{\ell}_\tau^{(t-1)}\right\|_{\Phi_{\eta,\gamma},p(\widehat{L}_t^{\mathrm{e}})} \\
&\leq 2 \sum_{\tau=t'}^{t-1} \left\|\hat{\ell}_\tau - \hat{\ell}_\tau^{(t-1)}\right\|_{\Phi_{\eta,\gamma},p(\widehat{L}_\tau^{\mathrm{e}})} \\
&\leq 2d_{\max}\sqrt{\gamma} \\
&\leq \frac{1}{16}.
\end{aligned}
$$

Thus, we can use Hölder's inequality to get:

$$
\begin{aligned}
&\left(p\left(\widetilde{L}_t\right) - p\left(\widehat{L}_t\right)\right) \cdot \ell_t \\
&\leq \left\|p\left(\widetilde{L}_t\right) - p\left(\widehat{L}_t\right)\right\|_{\Phi_{\eta,\gamma},p(\widehat{L}_t^{\mathrm{e}})}^* \|\ell_t\|_{\Phi_{\eta,\gamma},p(\widehat{L}_t^{\mathrm{e}})} \\
&\leq 8 \left\|\widetilde{L}_t - \widehat{L}_t\right\|_{\Phi_{\eta,\gamma},p(\widehat{L}_t^{\mathrm{e}})} \|\ell_t\|_{\Phi_{\eta,\gamma},p(\widehat{L}_t^{\mathrm{e}})} \\
&\leq 8 \left\|\sum_{\tau=1}^{t-1} \lambda_\tau^{(t-1)}\left(\ell_\tau - \hat{\ell}_\tau\right)\right\|_{\Phi_{\eta,\gamma},p(\widehat{L}_t^{\mathrm{e}})} \|\ell_t\|_{\Phi_{\eta,\gamma},p(\widehat{L}_t^{\mathrm{e}})} \\
&= 8 \left\|\sum_{\tau=t'}^{t-1} \lambda_\tau^{(t-1)}\left(\ell_\tau - \hat{\ell}_\tau\right)\right\|_{\Phi_{\eta,\gamma},p(\widehat{L}_t^{\mathrm{e}})} \|\ell_t\|_{\Phi_{\eta,\gamma},p(\widehat{L}_t^{\mathrm{e}})},
\end{aligned}
$$

where the second inequality is due to Fact 2 and last inequality is since $\lambda_\tau^{(t-1)} = 0$ for any $\tau < t'$.

Since our estimators are unbiased, $\mathbb{E}\left[\left(\ell_\tau - \hat{\ell}_\tau\right)\left(\ell_{\tau'} - \hat{\ell}_{\tau'}\right)\right] = 0$ for any $\tau \neq \tau'$, and thus

$$
\begin{aligned}
\mathbb{E}\left[\left\|\sum_{\tau=t'}^{t-1} \lambda_\tau^{(t-1)}\left(\ell_\tau - \hat{\ell}_\tau\right)\right\|_{\Phi_{\eta,\gamma},p(\widehat{L}_t^{\mathrm{e}})}^2\right] &= \sum_{\tau=t'}^{t-1} \left(\lambda_\tau^{(t-1)}\right)^2 \mathbb{E}\left[\left\|\ell_\tau - \hat{\ell}_\tau\right\|_{\Phi_{\eta,\gamma},p(\widehat{L}_t^{\mathrm{e}})}^2\right] \\
&= \sum_{\tau=t'}^{t-1} \left(\lambda_\tau^{(t-1)}\right)^2 \mathbb{E}\left[\left\|\hat{\ell}_\tau\right\|_{\Phi_{\eta,\gamma},p(\widehat{L}_t^{\mathrm{e}})}^2 - \|\ell_\tau\|_{\Phi_{\eta,\gamma},p(\widehat{L}_t^{\mathrm{e}})}^2\right] \\
&\leq \sum_{\tau=t'}^{t-1} \left(\lambda_\tau^{(t-1)}\right)^2 \mathbb{E}\left[\left\|\hat{\ell}_\tau\right\|_{\Phi_{\eta,\gamma},p(\widehat{L}_t^{\mathrm{e}})}^2\right] \\
&\leq 2 \sum_{\tau=t'}^{t-1} \left(\lambda_\tau^{(t-1)}\right)^2 \mathbb{E}\left[\left\|\hat{\ell}_\tau\right\|_{\Phi_{\eta,\gamma},p(\widehat{L}_\tau^{\mathrm{e}})}^2\right],
\end{aligned}
$$

where in the last step we used Fact 3 and Lemma 18 to move to the norms induced by $p\left(\widehat{L}_\tau^{\mathrm{e}}\right)$.

Using Eq. (8):

$$
\begin{aligned}
\mathbb{E}\left[\left\|\hat{\ell}_\tau\right\|_{\Phi_{\eta,\gamma},p\left(\widehat{L}_\tau^{\mathrm{e}}\right)}^2\right] &= \mathbb{E}\left[\sum_{i\in[K]} \frac{\eta\gamma p_i^2\left(\widehat{L}_\tau^{\mathrm{e}}\right)}{\eta + \gamma p_i\left(\widehat{L}_\tau^{\mathrm{e}}\right)} \frac{\ell_{\tau,i}^2 \mathbb{1}\left[i = a_\tau\right]}{p_i^2\left(\widehat{L}_\tau^{\mathrm{e}}\right)}\right] \\
&= \mathbb{E}\left[\sum_{i\in[K]} \frac{\eta\gamma p_i\left(\widehat{L}_\tau^{\mathrm{e}}\right)}{\eta + \gamma p_i\left(\widehat{L}_\tau^{\mathrm{e}}\right)} \ell_{\tau,i}^2\right] \\
&\le \eta K.
\end{aligned}
\tag{9}
$$

Combining the last equations and using Jensen's inequality, we thus have:

$$
\mathbb{E}\left[\left\|\sum_{\tau=t'}^{t-1} \lambda_\tau^{(t-1)}\left(\ell_\tau - \hat{\ell}_\tau\right)\right\|_{\Phi_{\eta,\gamma},p\left(\widehat{L}_t^{\mathrm{e}}\right)}\right] \le \sqrt{2\eta K \sum_{\tau=1}^{t-1}\left(\lambda_\tau^{(t-1)}\right)^2} \le \sqrt{2\eta K \lambda_t} \le \sqrt{\eta}(K + \lambda_t).
$$

Using Eq. (8) again, we have for any $\tau, t$:

$$
\|\ell_\tau\|_{\Phi_{\eta,\gamma},p\left(\widehat{L}_t^{\mathrm{e}}\right)}^2 = \sum_{i\in[K]} \frac{\eta\gamma p_i^2\left(\widehat{L}_t^{\mathrm{e}}\right)}{\eta + \gamma p_i\left(\widehat{L}_t^{\mathrm{e}}\right)} \ell_{\tau,i}^2 \le \sum_{i\in[K]} \eta p_i\left(\widehat{L}_t^{\mathrm{e}}\right) = \eta,
$$

so in total, we get

$$
H_3 = \sum_{t=1}^T \mathbb{E}\left[\left(p\left(\widetilde{L}_t\right) - p\left(\widehat{L}_t\right)\right) \cdot \ell_t\right] \le 8\eta\left(KT + \sum_{t=1}^T \lambda_t\right)
$$

as desired.

$H_2$.    For $H_2$, observe that the same as before, we have:

$$
\left\|\widetilde{L}_t - \widehat{L}_t^{\mathrm{e}}\right\|_{\Phi_{\eta,\gamma},p\left(\widehat{L}_t^{\mathrm{e}}\right)} \le \frac{1}{16} \qquad \text{and} \qquad \left\|\widetilde{L}_t^{\mathrm{e}} - \widehat{L}_t^{\mathrm{e}}\right\|_{\Phi_{\eta,\gamma},p\left(\widehat{L}_t^{\mathrm{e}}\right)} \le \frac{1}{16}.
$$

And thus we can use Hölder's inequality as before:

$$
\begin{aligned}
&\left( p\left( \widetilde{L}_t^{\mathrm{e}} \right) - p\left( \widetilde{L}_t \right) \right) \cdot \ell_t \\
&\leq 8 \left\| \widetilde{L}_t^{\mathrm{e}} - \widetilde{L}_t \right\|_{\Phi_{\eta,\gamma}, p\left( \widehat{L}_t^{\mathrm{e}} \right)} \left\| \ell_t \right\|_{\Phi_{\eta,\gamma}, p\left( \widehat{L}_t^{\mathrm{e}} \right)} \\
&\leq 8\sqrt{\eta} \sum_{\tau=1}^{t-1} \left( 1 - \lambda_\tau^{(t-1)} \right) \left\| \hat{\ell}_\tau^{(t-1)} - \hat{\ell}_\tau \right\|_{\Phi_{\eta,\gamma}, p\left( \widehat{L}_t^{\mathrm{e}} \right)} + 8\sqrt{\eta} \sum_{\tau=1}^{t-1} \lambda_\tau^{(t-1)} \left\| \ell_\tau^{(t-1)} - \ell_\tau \right\|_{\Phi_{\eta,\gamma}, p\left( \widehat{L}_t^{\mathrm{e}} \right)} \\
&\leq 8\sqrt{\eta} \sum_{\tau=1}^{t-1} \left( 1 - \lambda_\tau^{(t-1)} \right) \left\| \hat{\ell}_\tau^{(t-1)} - \hat{\ell}_\tau \right\|_{\Phi_{\eta,\gamma}, p\left( \widehat{L}_t^{\mathrm{e}} \right)} + 8\eta\lambda_t \\
&= 8\sqrt{\eta} \sum_{\tau=t'}^{t-1} \left( 1 - \lambda_\tau^{(t-1)} \right) \left\| \hat{\ell}_\tau^{(t-1)} - \hat{\ell}_\tau \right\|_{\Phi_{\eta,\gamma}, p\left( \widehat{L}_t^{\mathrm{e}} \right)} + 8\eta\lambda_t \\
&\leq 16\sqrt{\eta} \sum_{\tau=t'}^{t-1} \left( 1 - \lambda_\tau^{(t-1)} \right) \left\| \hat{\ell}_\tau^{(t-1)} - \hat{\ell}_\tau \right\|_{\Phi_{\eta,\gamma}, p\left( \widehat{L}_\tau^{\mathrm{e}} \right)} + 8\eta\lambda_t
\end{aligned}
$$

where again we denote $t' = \max\{1, t - d_{\max}\}$. The second inequality is due to Fact 2 and $\|\ell_t\|^2_{\Phi_{\eta,\gamma}, p\left( \widehat{L}_t^{\mathrm{e}} \right)} \leq \eta$, and the last inequality is due to Fact 3 and Lemma 18 to move to the norms induced by $p\left( \widehat{L}_\tau^{\mathrm{e}} \right)$.

Using Eq. (8):

$$
\begin{aligned}
\mathbb{E}\left[ \left\| \hat{\ell}_\tau^{(t-1)} - \hat{\ell}_\tau \right\|^2_{\Phi_{\eta,\gamma}, p\left( \widehat{L}_\tau^{\mathrm{e}} \right)} \right] &= \mathbb{E}\left[ \sum_{i \in [K]} \frac{\eta\gamma p_i^2\left( \widehat{L}_\tau^{\mathrm{e}} \right)}{\eta + \gamma p_i\left( \widehat{L}_\tau^{\mathrm{e}} \right)} \frac{\left( \ell_{\tau,i}^{(t-1)} - \ell_{\tau,i} \right)^2 \mathbb{1}\left[ i = a_\tau \right]}{p_i^2\left( \widehat{L}_\tau^{\mathrm{e}} \right)} \right] \\
&= \mathbb{E}\left[ \sum_{i \in [K]} \frac{\eta\gamma p_i\left( \widehat{L}_\tau^{\mathrm{e}} \right)}{\eta + \gamma p_i\left( \widehat{L}_\tau^{\mathrm{e}} \right)} \left( \ell_{\tau,i}^{(t-1)} - \ell_{\tau,i} \right)^2 \right] \\
&\leq \eta \sum_{i \in [K]} \left( \ell_{\tau,i}^{(t-1)} - \ell_{\tau,i} \right)^2 \\
&= \eta \left\| \ell_\tau^{(t-1)} - \ell_\tau \right\|^2_2,
\end{aligned}
$$

and so by Jensen's inequality:

$$
\mathbb{E}\left[ \left\| \hat{\ell}_\tau^{(t-1)} - \hat{\ell}_\tau \right\|_{\Phi_{\eta,\gamma}, p\left( \widehat{L}_\tau^{\mathrm{e}} \right)} \right] \leq \sqrt{\eta} \left\| \ell_\tau^{(t-1)} - \ell_\tau \right\|_2 = \sqrt{\eta} \frac{\lambda_\tau^{(t-1)}}{1 - \lambda_\tau^{(t-1)}}.
$$

Overall we get

$$
H_2 = \sum_{t=1}^{T} \mathbb{E}\left[ \left( p\left( \widetilde{L}_t^{\mathrm{e}} \right) - p\left( \widetilde{L}_t \right) \right) \cdot \ell_t \right] \leq 24\eta \sum_{t=1}^{T} \lambda_t.
$$

$H_4$. Note that:

$$\left\|\widehat{L}_t^* - \widehat{L}_t^{\mathrm{e}}\right\|_{\Phi_{\eta,\gamma},p\left(\widehat{L}_t^{\mathrm{e}}\right)} \leq \left\|\hat{\ell}_t\right\|_{\Phi_{\eta,\gamma},p\left(\widehat{L}_t^{\mathrm{e}}\right)} + \left\|\widehat{L}_t - \widehat{L}_t^{\mathrm{e}}\right\|_{\Phi_{\eta,\gamma},p\left(\widehat{L}_t^{\mathrm{e}}\right)} \leq (1 + 2d_{\max})\sqrt{\gamma} \leq \frac{1}{16},$$

Ss we can again use Fact 2:

$$\mathbb{E}\left[\left(p\left(\widehat{L}_t\right) - p\left(\widehat{L}_t^*\right)\right) \cdot \hat{\ell}_t\right] \leq 8\mathbb{E}\left[\left\|\widehat{L}_t - \widehat{L}_t^*\right\|_{\Phi_{\eta,\gamma},p\left(\widehat{L}_t^{\mathrm{e}}\right)} \left\|\hat{\ell}_t\right\|_{\Phi_{\eta,\gamma},p\left(\widehat{L}_t^{\mathrm{e}}\right)}\right]$$

$$= 8\mathbb{E}\left[\left\|\hat{\ell}_t\right\|_{\Phi_{\eta,\gamma},p\left(\widehat{L}_t^{\mathrm{e}}\right)}^2\right]$$

$$\leq 8\eta K$$

where the last step is due to Eq. (9). We can now complete the proof with:

$$H_4 = \sum_{t=1}^{T} \mathbb{E}\left[\left(p\left(\widehat{L}_t\right) - p\left(\widehat{L}_t^*\right)\right) \cdot \hat{\ell}_t\right] \leq 8\eta KT.$$

$\blacksquare$

## B.4. Skipping bound

**Lemma 20** *Denote the regret of some algorithm $\mathcal{A}$ compared to a loss sequence $\ell_1, \ldots, \ell_T$ with maximal delay $d_{\max}$ as $R_{d_{\max}}^{\mathcal{A}}\left(\{\ell_t\}_{1 \leq t \leq T}\right)$. When using Algorithm 3 with $\mathcal{A}$, the expected regret holds:*

$$R(T) \leq R_{d_{\max}}^{\mathcal{A}}\left(\left\{\ell_t^{(t+d_{\max})}\right\}_{1 \leq t \leq T}\right) + 2\sum_{t=1}^{T}\left\|\ell_t - \ell_t^{(t+d_{\max})}\right\|_{\infty}.$$

**Proof** We have:

$$R(T) = \max_{a \in [K]} \mathbb{E}\left[\sum_{t=1}^{T} \ell_{t,a_t} - \ell_{t,a}\right]$$

$$= \max_{a \in [K]} \mathbb{E}\left[\sum_{t=1}^{T}\left(\ell_{t,a_t}^{(t+d_{\max})} - \ell_{t,a}^{(t+d_{\max})}\right) + \left(\ell_{t,a_t} - \ell_{t,a_t}^{(t+d_{\max})}\right) + \left(\ell_{t,a}^{(t+d_{\max})} - \ell_{t,a}\right)\right]$$

$$\leq \max_{a \in [K]} \mathbb{E}\left[\sum_{t=1}^{T} \ell_{t,a_t}^{(t+d_{\max})} - \ell_{t,a}^{(t+d_{\max})}\right] + \sum_{t=1}^{T}\left\|\ell_t - \ell_t^{(t+d_{\max})}\right\|_{\infty}$$

$$= R_{d_{\max}}^{\mathcal{A}}\left(\left\{\ell_t^{(t+d_{\max})}\right\}_{1 \leq t \leq T}\right) + 2\sum_{t=1}^{T}\left\|\ell_t - \ell_t^{(t+d_{\max})}\right\|_{\infty},$$

where the last step is since algorithm $\mathcal{A}$ cannot distinguish between being wrapped in Algorithm 3 and $\left\{\ell_t^{(t+d_{\max})}\right\}_{1 \leq t \leq T}$ being the true losses with maximal delay $d_{\max}$. $\blacksquare$

