# OpenReview forum: "Non-stochastic Bandits With Evolving Observations"
_algorithmiclearningtheory.org/ALT/2025/Conference — ALT 2025_

### Official Review · Reviewer_DSiq · 2024-11-06
**New Model of Evolving Observations**

**Rating:** 8
**Confidence:** 3

**Review:**

**Summary**


The authors introduce a generalization of current notions of delayed rewards in adversarial bandits. The problem is motivated by online learning where the outcome of the action is not observed immediately, but the reward can be estimated. This estimation can evolve over time.

They propose and analyze algorithms for both the full information and bandit settings. Then, they demonstrate how commonly analyzed delay/corruption assumptions can be implemented in their framework.

**Pros**


-	The authors propose an interesting and useful generalization of adversarial delayed feedback. The bandit literature has many different assumptions on rewards and observations, and in my opinion, unifications like the ones proposed in this work are often helpful.
-	Standard algorithms can be adapted simply to this setting.
-	The result is related back to delayed, corrupted, and composite feedback so it can be compared to prior results.


**Cons**


-	The authors have some discussion of asymptotic optimality in the delayed setting, but it could include a more comprehensive discussion (e.g. is the algorithm optimal in the corrupted setting?)

**Paper Award:**

No

---

### Official Review · Reviewer_N33v · 2024-11-09
**The paper introduces a novel online learning framework with evolving feedback. Given the theoretical depth, the clarity of presentation, and the potential real-world applications, I believe this paper is a valuable contribution to the field of online learning and should be accepted for publication.**

**Rating:** 7
**Confidence:** 3

**Review:**

Summary:

The paper introduces a novel online learning framework that extends traditional delayed and corrupted feedback models to handle adversarial environments where feedback evolves over time. In this new setting, the observed feedback can change after each round, potentially overriding previous observations and introducing unique challenges for regret minimization.

The authors propose algorithms for both full-information and bandit settings, achieving regret bounds that depend on the accuracy of the feedback over time. This generalized evolving feedback setting includes many previous settings, like delayed feedback, corrupted feedback, optimistic delayed feedback, and composite delayed feedback. In the optimistic delayed feedback setting, after choosing the action, the agent observes the hints for the delayed feedback, which is also included in the evolving feedback setting. For this setting, they show the first regret bound for the bandit setting.

Pros:

(1) The paper successfully generalizes several existing feedback models (e.g., delayed, corrupted) into a unified framework, showing broad applicability to real-world problems.

(2) The paper is well-structured, and the explanations of both the model and algorithms are clear and accessible, making the theoretical contributions easy to follow.

(3) The regret analysis is thorough and demonstrates asymptotically optimal regret bounds for many previous settings including delayed feedback setting and composite delayed feedback setting. For the optimistic delayed feedback, they achieve the regret bound for the bandit setting which was previously not known.

Cons:

(1) Although the framework is extended to evolving feedback, the main algorithm (Algorithm 2) and its analysis share substantial similarities with the time-delayed feedback setting discussed in Van der Hoeven et al. (2023). This overlap may limit the perceived novelty of the core algorithmic contributions. Maybe the authors can elaborate more on the specific differences between the new regret analysis and that of Van der Hoeven et al. (2023), particularly in the aspects that handle evolving feedback.

(2) Section 3.2 is a little confusing. In Algorithm 3, there is an otherwise part in the last line, which is not clear in which case to use and seems not required since the part before it already truncated at $\tau+d_{\max}$. In Corollary 10, the $\bar{\Lambda}$ is an upper bound on the total inaccuracy (ignoring inaccuracy larger than $d_{\max}$), but ignoring the large delay part is not shown in the formula. Also, since here you assume the feedback for the large delays is $\varepsilon$ accurate, it seems unnecessary to include this.
Finally, the regret bound in Section 3.1 works for the maximum delay is small and the regret bound in Section 3.2 can handle the large delay with the assumption that the feedback is fairly accurate for large delays. What would the regret bound be for the large delay without this assumption? Is the proposed regret bound still working?

**Paper Award:**

No

---

### Official Review · Reviewer_CVnL · 2024-11-16
**Analysis of regret when loss observations change over time**

**Rating:** 6
**Confidence:** 3

**Review:**

This paper considers bandit problems where the feedback over rewards is delayed (or evolving over time).

The regret of their algorithms achieve depends on the time averaged (over both rounds and prior rounds’) $\ell_2$ distance between the feedback loss and the true loss.   The algorithms they analyze exponential weights in the full info setting, and a variant of FTRL adapted to this setting. While the analysis is reasonably straightforward, this reviewer agrees that the results are a nice way to generalize the general bandit feedback framework. As someone who is not an expert in results in related delayed or imperfect feedback settings, the reviewer will defer to others more familiar with the relevant literature to the novelty of these results.

Minor:

The reviewer would like a more detailed explanation of what is similar and different between this work and (van der Hoeven et al., 2023).

The related work feels rushed and could be more thoughtful in its treatment.

**Paper Award:**

No

---

### Author Rebuttal · Authors · 2024-11-18

Thank you all for your helpful feedback. To address specific concerns:

Reviewers CVnL and N33v, in addition to extending the model of (van der Hoeven et al., 2023), our analysis is different (although admittedly similar in structure) - we use feedback accuracy measures as a way to have a continuous domain of delays (instead of a binary delay/no-delay as in their work). This introduces two extra drift terms to bound that require a careful choice of the accuracy measures.

As for the second point of reviewer N33v, the last line of algorithm 3 is indeed a bit confusing - we mean there that for large delays, the wrapper should return the previously seen feedback. This is an artificial requirement since in our model the agent always gets new observations (which may just be the same as previous ones). As for Corollary 10, it’s necessary to ignore the large delays while computing $\Lambda$, since otherwise we get a $\sqrt{\varepsilon}T$ term (this will become clear when we add the explicit formula, which shows $\Lambda \approx \varepsilon T^2$). Finally, we have no assumptions on $\varepsilon$, so you can use this result when there are inaccurate large delays (in which case the regret bound might be in the order of $T$ which is not very helpful, but that’s all you can get without accurate feedback).

For reviewer DSiq, our results (algorithms 2+3) are optimal on all the special cases we show (and know of). That said, we have no proof they are optimal in the general case, and we leave as an open question to find a matching lower bound.

We will try to address all of your concerns in the final version of the paper. Thanks again for the reviews.

---

> ### Comment · Reviewer_N33v · 2024-11-20
>
> Thanks authors for the detailed responses, which address all my questions. I will maintain my score.

---

### Meta-Review · Area_Chair_Hebv · 2024-12-05

**Recommendation:** Accept
**Confidence:** 4

**Metareview:**

This work considers the experts problem given bandit feedback in a setting where the past loss functions evolve (and hopefully approach the true loss eventually) rather than staying fixed. In doing so, it provides a clean and natural generalization of related settings where the loss functions are observed with (non-uniform) delays and where the feedback is corrupted for a few rounds. Beyond the algorithm and the analysis, this is a conceptual contribution in itself.

The paper was positively received by the reviewers. Further, in the rebuttal, the authors describe differences in analysis (handing non-binary drift terms) with respect to previous work on loss-with-delay settings.

Happy to recommend this paper for acceptance.

**Paper Award:**

No